# Sperm as a Carrier of Genome Instability in Relation to Paternal Lifestyle and Nutritional Conditions

**DOI:** 10.3390/nu14153155

**Published:** 2022-07-30

**Authors:** Usha Punjabi, Ilse Goovaerts, Kris Peeters, Helga Van Mulders, Diane De Neubourg

**Affiliations:** 1Centre for Reproductive Medicine, Antwerp University Hospital, 2650 Edegem, Belgium; ilse.goovaerts@uza.be (I.G.); kris.peeters@uza.be (K.P.); helga.vanmulders@uza.be (H.V.M.); diane.deneubourg@uza.be (D.D.N.); 2Department of Reproductive Medicine, Antwerp Surgical Training, Anatomy and Research Centre (ASTARC), Faculty of Medicine and Health Sciences, University of Antwerp, 2000 Antwerpen, Belgium

**Keywords:** sperm DNA fragmentation, semen parameters, chromatin maturity, oxidative stress, sperm aneuploidy, genome instability, lifestyle factors, male age, male BMI

## Abstract

Endogenous and exogenous factors can severely affect the integrity of genetic information by inducing DNA damage and impairing genome stability. The extent to which men with and without subfertility are exposed to several adverse lifestyle factors and the impact on sperm DNA fragmentation (SDF), sperm chromatin maturity (condensation and decondensation), stability (hypo- and hypercondensation) and sperm aneuploidy are assessed in this study. Standardized assays employing flow cytometry were used to detect genome instability in 556 samples. Semen parameters deteriorated with age, BMI, increased physical activity and smoking. Age and BMI were associated with increased SDF. Increased BMI was associated with increased hypocondensed chromatin and decreased decondensed chromatin. Increase in age also caused an increase in sex chromosome aneuploidy in sperms. Surprisingly, alcohol abuse reduced chromatin hypercondensation and drug abuse reduced SDF. Although genome instability was more pronounced in the subfertile population as compared to the fertile group, the proportion of men with at least one lifestyle risk factor was the same in both the fertile and subfertile groups. While one in three benefited from nutritional supplementation, one in five showed an increase in SDF after supplementation. Whilst the message of ‘no smoking, no alcohol, no drugs, but a healthy diet’ should be offered as good health advice, we are a long way from concluding that nutritional supplementation would be beneficial for male fertility.

## 1. Introduction

Infertility affects nearly 15% of couples [1], and a male factor contributes in up to 50% of these cases [2]. A temporal trend in semen quality has been observed, giving evidence for a decreasing quality of semen in the past 50 years [3]. Recently, it has been indicated that impaired semen quality is associated with shorter life expectancy and increased long-term morbidity [4,5,6,7], emphasizing the significance of diagnosing male infertility.

Semen analysis is considered as the cornerstone of male fertility evaluation, providing fundamental information on which clinicians base their initial diagnosis. However, semen analysis is considered to be subjective, poorly standardized and insufficient as a predictor [2,8,9,10,11]. A substantial overlap of semen parameters between fertile and infertile males has been reported [11]. Nevertheless, fertility is not only based on the absolute numbers of spermatozoa but also on their functional capability. Multiple technologies exploring chromatin structure anomalies have been applied during the last decade [12,13,14,15,16] to evaluate fertility disorders and to increase the predictive value of sperm analysis for procreation in vivo and in vitro [17]. DNA/chromatin integrity in sperm has been by far the most studied molecular feature in sperm and has been linked with a wide variety of pathological outcomes, including reduced fertilization rates, impaired preimplantation development, and an increased incidence of miscarriage and morbidity in offspring [18,19,20,21,22,23,24,25].

Genome stability is a feature of every organism to preserve and faithfully transmit genetic material from generation to generation. Endogenous and exogenous factors can severely affect the integrity of genetic information by inducing DNA damage and impairing genome stability. DNA damage is the most important factor that induces genome instability. When DNA repair processes fail, irreparable DNA damage including single and double-strand breaks can occur. Defective spermatogenesis and abnormalities in chromatin remodeling and abortive apoptosis are the major factors affecting the integrity of testicular sperm DNA, while testicular and post-testicular oxidative stress (OS) might also induce DNA damage [26,27].

During spermiogenesis, a rearrangement of the cytoskeletal structure transforms round spermatids into mature spermatozoa. Histones are replaced by transition proteins, and then by protamines (P1 and P2). This transition is associated with an occurrence of DNA strand breaks necessary for the transient relief of torsional stress, favoring casting off the nucleosome histone cores and aiding their replacement with protamines [28]. Chromatin packaging also requires endogenous nuclease activity to loosen chromatin by histone hyper-acetylation and the introduction of breaks with topoisomerase II, which is capable of both creating and ligating breaks. These combined DNA-condensing activities may optimize the strand repair process, emphasizing the link between altered sperm DNA condensation and DNA fragmentation [29]. Abnormally high amounts of histones in sperm are associated with decreased fertility and an increased risk of embryonic failure after fertilization [30]. Therefore, histone retention and protamine deficiency in sperm are hallmarks of certain forms of idiopathic infertility [31,32,33,34]. During the sperm passage in the epididymis, cross-linking between cysteine residues of protamine disulfide bonds enhances the stabilization of the nucleoprotamine complex [35,36]. 

Conversely, during fertilization, healthy spermatozoa must decondense and reorganize into nucleosomal structures. The protamines are removed from the DNA by a process in which disulphide bonds are reduced before binding the polycationic protamine to nucleoplasmin, a small negatively charged protein in the egg [37]. When this happens, the sperm nucleus decondenses and the DNA combines with egg histones, forming the male pronucleus. Defects of sperm chromatin that prevent or delay chromatin decondensation can be expected to prevent the normal development of the male pronucleus. The oocyte has an important DNA repair capacity, but is largely efficient in relation to DNA strand breaks [38]. The oocyte capacity to repair sperm maturity or stability defects is rather limited. Therefore, the determination and correction (when possible) of decondensation is of paramount importance in assisted reproductive technologies [39].

However, many men have no known cause of infertility (idiopathic). Chromosomal aberrations, either numerical or structural, can have profound effects on fertility [40,41]. Infertile males produce gametes with a higher rate of chromosomal abnormalities than those found in the general population [42]. Chromosome stability is of crucial significance in cell division and propagation. An abnormal number of chromosome(s) during unbalanced cell separation at cell division is associated with almost all solid tumor cancers. 

Male infertility is a multifactorial disease that can be caused by a wide variety of genetic and acquired lifestyle factors. Adverse health behaviors such as excessive alcohol intake, smoking, recreational drugs and obesity are associated with reduced fertility in men [43,44]. The amelioration of lifestyle factors may improve semen parameters but reports on the effects on sperm genome stability via sperm DNA integrity and chromatin maturity are scarce. Such data may have important implications in male subfertility, and the effectiveness of treatment given to couples with male subfertility.

This study is undertaken to assess which lifestyle parameters influence sperm genome stability and whether this influence on semen quality is mediated via OS. Moreover, if OS plays a role in sperm dysfunction, antioxidant therapy would appear a logical approach. Can nutrient supplementation be beneficial for male infertility?

## 2. Materials and Methods

### 2.1. Study Protocol

The study was monocentric, cross-sectional, prospective and partly retrospective. Recruitment and data collection occurred during different phases. Accordingly, the Ethical Commission of the Antwerp University Hospital and the University of Antwerp approved several projects. 

Sperm DNA fragmentation in a fertile population conducted between October 2017 and October 2020, approved on 26 June 2017, ref. no: 17/24/285 (Belgian registration no: B300201732872);Sperm DNA fragmentation in an infertile population conducted between October 2017 and October 2020, approved on 11 August 2017 (Belgian registration no: B300201733352).OS conducted between January 2017 and March 2018, approved on 31 July 2017, ref. no: 17/29/321 (Belgian registration no: B300201733042).Chromatin maturity and stability conducted between January 2020 and March 2020, approved on 13 January 2020, ref. no: 19/51/629.Sperm aneuploidy retrospective data collection between January 2014 and December 2016, approved on 6 July 2020, ref. no: 20/26/350.

### 2.2. Participants 

The study population comprised a cohort of patients (18–65 years old) undergoing their first infertility diagnosis and treatment at the Centre for Reproductive Medicine, Antwerp University Hospital, Belgium. Subjects were excluded on the following terms: azoospermia (no spermatozoa) and cryptozoospermia (few hidden spermatozoa). Another group of fertile men (who achieved pregnancy within 12 months of unprotected coitus) and sperm donors (who had self-fathered children or had achieved pregnancies within the donor program of the clinic) were included as a control group. All subjects had given written informed consent for participation.

### 2.3. Procedure and Intervention

All participants filled in a clinical male fertility diagnosis questionnaire as part of a standard diagnostic procedure covering personal medical history, clinical pathologies (such as varicocele, cryptorchidism, testicular infection, prostatitis and testicular torsion) and lifestyle parameters (smoking, alcohol use and drug abuse). Body mass index (BMI) was assessed by the patient’s current height and weight. To measure physical activity a second validated questionnaire was requested [45] covering three different categories of activities: occupation, sports and leisure time. All stated questions were pre-coded on a five-point scale while the main occupation and the types of sport practiced were scored on a three-point scale. The higher the intensity and time expenditure, the higher the physical activity score. 

An exploratory approach was initiated in a randomly selected group of infertile men with a nutritional support (Condensyl TM) of fig fruit extract (Opuntia ficusindica, 100 mg), quercetin (0.001 mg), betalain (0.05 mg) and a mix of Group B vitamins (B2 (1.4 mg), B3 (16 mg), B6 (1.4 mg), B9 (400 µg), B12 (2.5 µg)), together with zinc (12.5 mg), L-cysteine (170 mg) and Vitamin E (12 mg). The prescribed dose was 1 tablet/day with a targeted treatment duration of 3–4 months to overcome one spermatogenic cycle. Semen samples were analyzed before and after intervention.

### 2.4. Semen Analysis

Semen samples were collected at the laboratory and the analysis initiated within 60 min of ejaculation to conform to the international standards of ISO 15189 (International Standards Organization, 2012). Standard semen parameters including sperm concentration, motility and morphology were determined using WHO 2010 [46] recommendations, complying with the checklist for acceptability reported by Björndahl et al. [47]. All staff members were trained in basic semen analysis (ESHRE—European Society for Human Reproduction and Embryology Basic Semen Analysis Courses) [48,49] and participated regularly in internal and external quality control programs (Institute of Public Health, Belgium and ESHRE External Quality Control Schemes, Stockholm, Sweden) [50].

### 2.5. Oxidative Stress (OS)

Oxidation-reduction potential (ORP) is a direct measurement of OS in semen samples and requires the measurement of the transfer of electrons from an antioxidant or reductant to an oxidant. In this way, the existing balance between total oxidants and reductants is measured. The Male Infertility Oxidative System (MiOXSYS) is a galvano stat-based technique that measures this balance [51]. The device consists of an analyzer and a sensor strip. Before testing, the sensor is pre-inserted into the MiOXSYS analyzer. Subsequently, 30 μL of the liquefied semen sample is applied to the sensor using a pipette. When the measurement is completed, ORP is displayed in millivolts (mV). The value is normalized by dividing it with sperm concentration to control for differences in cell numbers and data are presented as mV/M/mL semen. A cut-off value of 1.94 mV/M/mL could distinguish between normal and subnormal semen samples.

### 2.6. Sperm DNA Fragmentation (SDF)

An assessment of SDF was performed using terminal deoxynucleotidyl transferase-mediated deoxyuridine triphosphate nick-end labeling (TUNEL assay) described by Mitchell et al. [52]. Briefly, spermatozoa were incubated for 30 min at 37 °C with LIVE/DEAD^®^ Fixable Dead Cell Stain (far red) (Molecular Probes, Life technologies, Eugene, Oregon, USA), after which the cells were washed twice with phosphate-buffered saline (PBS, GIBCO Life technologies, Paisley, UK) before being incubated with 2 mM dithiothreitol (DTT, Sigma-Aldrich, Overijse, Belgium) for 45 min. Following this, the samples were washed 2 times in PBS and fixed in 3.7% formaldehyde (Sigma-Aldrich, Belgium) for 20 min at 4 °C. As storage of the sample at 4 °C affects reproducibility [9], the assay was carried out directly on fresh semen samples without storage. For the assay, the spermatozoa were washed twice and centrifuged before being resuspended in 500 µL of fresh permeabilization solution (100 mg Sodium citrate, 100 µL Triton X–100 in 100 mL dH2O) and incubated for 5 min at 4 °C. The cells were washed twice with PBS. The positive control samples were treated with 5 µL of DNase I (Qiagen, Hilden, Germany) 1500 Kunitz Units for 30 min at room temperature. The assay was performed using the fluorescein In Situ Cell Death Detection Kit (Roche Diagnostics, Mannheim, Germany) using an Accuri C6 flow cytometer (BD Sciences, Erembodegem, Belgium). For each sample, 5000–10,000 events were recorded at a flow rate of 35 µL/min.

DNA fragmentation was analyzed in the total sperm sample (total SDF) comprising viable and nonviable sperms, as well as in the vital fraction (vital SDF), thereby analyzing only viable sperm. The method was standardized and cut-off values were defined [53,54].

### 2.7. Sperm Nuclear Chromatin Condensation and Decondensation Assessment

Sperm chromatin condensation and decondensation were evaluated according to the procedure by Molina et al. [55]. In brief, semen samples were aliquoted into two fractions. The first aliquot was treated with the DNA-intercalating dye propidium iodide (Sigma-Aldrich, Overijse, Belgium, PI, 50 µg/mL) followed by a flow cytometric evaluation of the PI fluorescence intensity on a cell per cell basis. This was carried out on a Facscan (BD Biosciences, Erembodegem, Belgium) equipped with standard excitation and emission optics. The resulting PI fluorescence frequency distribution reflected the status of DNA condensation in the measured nuclei. The second aliquot was treated with 1% sodium dodecyl sulphate (SDS, Sigma-Aldrich, Overijse, Belgium) plus 6 mmol/l ethylene diamine tetra acetic acid (EDTA, Sigma-Aldrich, Overijse, Belgium)-decondensing solution in borate buffer (Sigma-Aldrich, Overijse, Belgium) for 5 min before using PI. Approximately 3000–9000 cells for each sample were analyzed. The mean channel of fluorescence was used to analyze the accessibility and, consequently, the degree of staining of sperm DNA with PI and the following flow cytometry parameters were analyzed:Condensed chromatin—histones replaced by protamines, transforming the nucleus into a highly compact structure;Hypocondensed chromatin—insufficient chromatin condensation or a potential condition of underprotamination rendering the paternal genome susceptible to damage;Decondensed chromatin ability of compacted chromatin to decondense in vitro after sodium dodecyl sulphate (SDS) + EDTA treatment;Hypercondensed chromatin—resistance to decondensation achieving a state of hyperstability making the paternal genome unavailable for further fertilization.

The method was standardized and cut-off values defined for all chromatin parameters [56].

### 2.8. Fluorescence In Situ Hybridization (FISH) Analysis

Sperm samples were washed with phosphate-buffered saline (PBS; Gibco; Life Technologies, Paisley, UK) and the resulting pellet fixed in Carnoy’s solution (methanol/acetic acid, 3:1; Merck, Overijse, Belgium). The fixed specimens were stored at −20 °C until further processing. Cytogenetic analysis of 5 chromosomes: chromosomes 13, 18, 21, and X/Y were performed according to Vegetti et al. [57]. Briefly, the fixed spermatozoa were spread on slides and air-dried. For nuclear decondensation, the air-dried slides were washed twice with saline citrate solution (20X SSC, Invitrogen, Merelbeke, Belgium) and incubated in 1 mol/L Tris buffer containing 25 mmol/L dithiothreitol (DTT, Sigma-Aldrich, Overijse, Belgium). Following decondensation, the slides were washed twice with SSC and dehydrated through an ethanol series and air-dried. A two-color FISH using locus-specific probes for chromosomes 13 (spectrum green) and 21 (spectrum red) and a three-color FISH with centromeric probes for chromosomes X (spectrum green), Y (spectrum red) and 18 (spectrum blue) was performed. Vysis (Abbott Laboratories) supplied the probes and the FISH protocol performed according to Vysis. Slides were observed using an Axioplan epifluorescence microscope (Leica, Wetzlar, Germany) with appropriate filter sets. For each probe, a maximum of 1000 spermatozoa were counted per patient. Only intact spermatozoa with clear hybridization signals were scored, and disrupted or overlapping spermatozoa were excluded. Sperm nuclei were scored nullisomic when no signals for the investigated chromosomes were seen. Sperm nuclei were considered disomic when two similar signals of the same color were observed. Finally, sperm nuclei were considered as diploid when two signals for each tested chromosome were exhibited in an intact spermatozoa. WHO 2021 [58] values for sperm disomy and our own fertile population levels for nullisomy were adapted.

With the causative factors analyzed, methods for genome instability assessment as well as the consequences are schematically summarized in Figure 1.

### 2.9. Statistical Analysis

Statistical analyses were conducted using Medcalc^®^ version 13.0.6.0 (MedCalc Software Bv, Oostende, Belgium) and IBM SPSS statistics version 26.0. 2019, Armonk, NY, USA. Descriptive statistics (mean, standard deviation (SD) and range) are reported for the patient characteristics, semen parameters, SDF parameters, chromatin parameters, and chromosome aneuploidy. Spearman correlation was calculated between SDF parameters, chromatin maturity and stability, sperm aneuploidy, and patient characteristics and semen parameters.

Semen variables where necessary were back transformed after logarithmic transformation. Data distributions were evaluated by the Kolmogorov–Smirnov test. The unpaired Student’s t-test was used in cases of normal distribution and the Mann–Whitney test was used in cases where the data were not normally distributed. Differences in continuous variables between 3 or more groups were assessed using the ANOVA and Kruskal–Wallis tests [59].

With the exception of chromatin condensation data, where an unpaired Student’s t-test was conducted, all other chromatin, semen and SDF parameters “rejected normality” (*p* < 0.05) for which the Mann–Whitney test was used to assess differences in continuous variables between two groups. Differences in continuous variables between 3 or more groups were assessed using the ANOVA and Kruskal–Wallis tests. If significant, the groups were compared pairwise using a post hoc test. Comparisons of the data distributions between the fertile and subfertile groups were conducted by constructing receiver operating characteristic (ROC) curve analysis. For all statistical tests, differences with a *p* value < 0.05 were considered significant.

## 3. Results

Semen parameters were assessed in 556 samples (fertile and subfertile). In 315 samples (56.6%), semen parameters were normal while one or more abnormalities were noted in the rest. SDF parameters were available in 547 and ORP in 241 samples, respectively. Maturity was determined in an additional 75 chromatins, and sperm aneuploidy was determined in another 223 samples. Descriptive characteristics of all participants are given in Table 1.

### 3.1. Lifestyle Factors Affecting Sperm Genome Instability in the Infertile Group

Age: With increasing age, semen parameters deteriorated significantly (concentration *p* = 0.030; total count *p* = 0.026; progressive motility *p* ≤ 0.001; total motility *p* = 0.001; morphology *p* = 0.002) (Figure 2). Total SDF increased (*p* < 0.001) without affecting vital SDF (*p* = 0.309). ORP was not associated with increasing age.

Age was not significantly associated with chromatin condensation (r = 0.09, *p* = 0.490), decondensation (r = 0.05, *p* = 0.693), hypocondensation (r = 0.09, *p* = 0.448) or hypercondensation (r = 0.19, *p* = 0.137).

On the other hand, paternal age was significantly correlated with the nullisomy of chromosomes 13 (r = 0.15; *p* = 0.023), 18 (r = 0.15; *p* = 0.031) and gonosomes (r = 0.19; *p* = 0.005). Sex aneuploidy gave a positive significant correlation (r = 0.14; *p* = 0.038) but not autosomal aneuploidy or diploidy.

BMI: An increase in BMI reduced total sperm motility (*p* = 0.022), but not the other semen parameters. Total SDF increased (*p* = 0.009) (Figure 3) without affecting vital SDF (*p* = 0.169). ORP was not associated with increasing BMI. Age positively correlated with BMI (*p* < 0.001). Classifying the weight status by BMI: 12 men (3.2%) were underweight (<18.5 kg/m^2^); 197 (52.7%) had a normal weight (18.5–25 kg/m^2^); 132 (35.3%) were overweight (25–30 kg/m^2^); and 33 (8.8%) were obese (>30 kg/m^2^). There was no significance (*p* = 0.1501) observed in total SDF between the different categories.

Chromatin parameters were not affected by BMI (condensed: r = 0.15, *p* = 0.424; decondensed: r = 0.31, *p* = 0.097; hypocondensed: r = 0.17, *p* = 0.369; hypercondensed: r = 0.29, *p* = 0.127, respectively). Grouping the men in the different BMI categories gave a significant difference in decondensation (*p* = 0.047), with post hoc analysis revealing a significant difference between the group with normal weight (78.9 ± 9.9%) and the obese group (60.6 ± 22.7%). A significant difference in hypercondensation was also observed (*p* = 0.026) with the post hoc test revealing a significant difference between the normal weight (5.2 ± 3.5%) and the overweight (13.7 ± 9.7%) and obesity (14.1 ± 6.8%) groups. No statistically significant difference in condensation (*p* = 0.244) or hypocondensation (*p* = 0.291) between the groups was found.

Physical activity: Sperm concentration and motility were not associated with physical activity (concentration: r = 0.10, *p* = 0.198; total count: r = 0.04, *p* = 0.588; progressive motility: r = 0.01, *p* = 0.887; total motility: r = 0.03, *p* = 0.731). Sperm morphology, on the other hand, significantly (r = 0.18, *p* = 0.016) reduced with increased physical activity (Figure 4). ORP (r = 0.03, *p* = 0.469), total SDF (r = 0.12, *p* = 0.120) and vital SDF (r = 0.11, *p* = 0.178) were not associated with physical activity.

Smoking: Out of the 95/400 (23.7%) patients reported to be smoking, six had a past history of abuse and four other candidates were using e-cigarettes and an oriental tobacco pipe. Smoking significantly reduced concentration (74.3 ± 70.0 vs. 60.2 ± 60.0 M/mL; *p* = 0.009) and morphology (5.5 ± 3.6 vs. 4.8 ± 3.7%; *p* = 0.025) but not motility (55.2 ± 15.5 vs. 58.4 ± 13.5%; *p* = 0.115), as compared to non-smokers. The group of smokers were significantly older (36.8 ± 7.4 vs. 32.9 ± 8.8 years; *p* < 0.001) with an increased BMI (26.0 ± 3.9 vs. 24.7 ± 3.8; *p* = 0.009), as compared to the non-smokers. ORP (2.1 ± 3.5 vs. 2.6 ± 7.6 mV/M/mL; *p* = 0.085), total SDF (9.9 ± 7.8 vs. 10.5 ± 8.7%; *p* = 0.338) and vital SDF (1.1 ± 1.2 vs. 1.3 ± 1.6%; *p* = 0.459) were not affected by smoking.

Categorizing smokers into current and former users revealed a significant difference in age, BMI and sperm morphology (Table 2). The post hoc test comparing the three groups pairwise showed a significant effect between non-smokers and current smokers for age (*p* = 0.001), BMI (*p* = 0.035) and morphology (*p* = 0.047).

When categorizing smokers according to the number of cigarettes used/day into light (<10 cigarettes; *n* = 27), moderate (10–20 cigarettes; *n* = 34) and heavy (>20 cigarettes; *n* = 20) smokers, only age revealed significance (*p* = 0.001) (Figure 5). The post hoc test gave a significant effect of age between non-smokers and light smokers (*p* = 0.008) and between non- and heavy smokers (*p* = 0.007). The participants were significantly older in the heavy smokers’ group.

Alcohol: 290/405 (71.6%) reported consuming alcohol. One was reported as a former alcohol abuser. Semen parameters were not significantly different in the group consuming alcohol compared to abstainers (concentration: 68.1 ± 62.6 vs. 79.4 ± 78.0 M/mL, *p* = 0.187; progressive motility: 50.3 ± 14.5 vs. 52.0 ± 11.4%, *p* = 0.407 and morphology: 5.4 ± 3.6 vs. 5.4 ± 3.8%, *p* = 0.725). Alcohol consumption did not affect total SDF, vital SDF, semen parameters or ORP significantly. Alcohol consumers were slightly younger (33.1 ± 8.6 vs. 34.7 ± 9.0; *p* = 0.071) with a significantly lower BMI (24.7 ± 3.8 vs. 25.4 ± 3.9; *p* = 0.024). Although alcohol consumption varied between light (<10 units/week, *n* = 213), moderate (10–20 units/week, *n* = 56) and heavy (>20 units/week, *n* = 7) drinking, the degree of consumption did not affect SDF total (*p* = 0.079). Alcohol consumption was strongly associated with hypercondensed chromatin (r = 0.64; *p* = 0.001). Alcohol consumers (3/22) had a significantly lower percentage of hypercondensed chromatin (9.8 ± 5.1 vs. 21.4 ± 4.7%; *p* = 0.015).

Drugs: Only 54/404 (13.4%) reported having used drugs. Drug abuse did not affect the semen parameters or ORP significantly. The use of drugs apparently reduced total SDF significantly (10.6 ± 8.3 vs. 9.4 ± 9.4; *p* = 0.043) without affecting the vital SDF (*p* = 0.697). The different types of drugs used showed no effect on total SDF (Figure 6).

In the small number of cases using drugs no effect was observed on chromatin maturity or stability.

In order not to miss important covariates, multiple regression analysis was carried out as an objective approach to analyze the relationship between SDF and all lifestyle factors (Table 3). Total SDF was significantly associated only with age, and vital SDF showed no significance with any lifestyle factors.

The different chromatin parameters revealed no significant association with any lifestyle factors. ORP, on the other hand, was positively and significantly associated with vital SDF (Table 4).

### 3.2. Lifestyle Factors Affecting Sperm Genome Instability in a Fertile Group

There was no significant correlation of SDF with lifestyle parameters in the fertile control group analyzed, except for a weak positive association of total DNA with BMI (*p* = 0.055). The fertile group tended to be younger (31.3 ± 6.1 vs. 34.0 ± 9.0 years; *p* = 0.068) and had significantly better semen parameters (Table 5). Fertile men smoked less (9.5% vs. 23.7%; *p* = 0.036) and used drugs less (7.1% vs. 13.4%; *p* = 0.245) but consumed an equal amount of alcohol (69.0% vs. 71.6%; *p*= 0.723). There was no significant difference in total or vital SDF. Chromatin condensation, decondensation and hypercondensation parameters scored significantly better in the fertile group. There was no significant difference in the hypocondensed population, although the wide ranges present in the subfertile group denote the susceptibility of chromatin to potential damage. In the fertile population, the autosomal aneuploidy and diploidy were significantly lower than in the subfertile group, while sex aneuploidy revealed no significant difference. Multiple regression analyses failed to reveal any association between lifestyle factors and SDF in the fertile population.

### 3.3. Incidence of Lifestyle Risk Factors in the Fertile and Subfertile Groups

Comparison of age was conducted by constructing ROC curve analysis (sensitivity 95.5%, specificity 21.6%, *p* = 0.0366) and threshold criteria were determined using the Youden J index (≤40 years). Taking into consideration all five lifestyle parameters analyzed, we could define a healthy lifestyle as: ≤40 years age, normal weight category (18.5–25 kg/m^2^) for BMI, non-smoker, alcohol abstainer and drug abstainer.

Approximately, 60.5% of the men under investigation for subfertility in our study had at least one lifestyle risk factor, 28.3% had two or more lifestyle risk factors, and only 11.3% had a healthy lifestyle. The healthy lifestyle was better but not significantly different in the fertile group (19.6%); 63.0% of them had at least one risk factor and 17.4% had two or more.

### 3.4. Nutritional Intervention and Genome Instability in the Infertile Group

Out of the 25 patients who gave consent for nutritional intervention, 23 patients completed the intervention and brought in semen samples for analysis. One sample proved insufficient for further analysis. Five had normozoospermia, while the rest had one or more semen abnormalities (oligozoospermia and/or asthenozoospermia and/or teratozoospermia). Semen and SDF parameters are summarized in Table 6.

In 47.4% of the semen samples total SDF was high (>13%) before intervention [54]. Only one in three benefited from this oral supplementation. In the rest (52.6%), total SDF was well within the fertile levels (≤13%), and a greater percentage retained their low levels. One in five, on the contrary, showed an increase in SDF after supplementation. Vital SDF was high (≥2%) in 26.3% of the semen samples and 60% of these benefited from supplementation. Those with a low vital SDF retained their low levels even after intervention (Figure 7).

## 4. Discussion

In the absence of any effective pharmacological intervention for declining male infertility, correct lifestyle advice remains a critical aspect of treatment for couples with male subfertility. In this study, we report for the first time the extent to which men with and without subfertility are exposed to several adverse lifestyle factors and the impact this has on sperm genome stability, which was more compromised in the subfertile than in the fertile group.

It has been postulated that fertile men with normal semen parameters have almost uniformly low levels of DNA breakage, whereas infertile men, especially those with compromised semen parameters, have increased proportions of nicks and breaks in the chromatin [60,61]. In our previous study [53], no differences in the levels of total SDF were observed between normal and subnormal samples, suggesting that sperm DNA damage may be one of the factors related to unexplained male infertility, especially in normozoospermia [62,63]. Evidently, SDF may be considered as an independent attribute of semen quality for all infertility patients, detecting problems not seen with semen analysis alone. Oxidative stress is stated to be one of the major contributory factors to DNA damage. The vulnerability of the spermatozoa to free radical attack and the induction of the lipid peroxidation process disrupt the integrity of the plasma membrane and impair sperm motility [63,64,65,66,67]. We found that vital SDF was significantly associated with ORP. The fact that no correlation was found between ORP and total SDF is in line with the recent review by Caroppo and Datillo, stating that double-strand breaks are mainly associated with defective histone to protamine transition, and not with oxidative damage [68]. Moreover, sperm chromatin parameters reveal a low correlation with standard semen parameters, suggesting that these reflect completely different physiological processes during spermatogenesis [55]. Sperm nuclear maturity and chromatin stability appear to be more homogenous in a fertile population and heterogeneous in a patient population. Incorrect chromatin compaction exposes spermatozoa to DNA damage [69]. However, abnormal sperm chromatin packaging can also be manifested as a supernormal compaction which would prevent the delivery of the male genome in the oocyte [70]. Any abnormalities in the unique organization of sperm chromatin are thought to affect the proper expression and regulation of paternal genes in the early embryo [71].

Although sperm cells provide half of the nuclear DNA, genetic alterations contributing to male factor infertility have been restricted to karyotypes. Sperm aneuploidy frequencies are largely consistent over time in men with proven fertility with little intra-individual variation [72]; however, high frequencies are reported in infertile males [73,74,75,76]. Moreover, the WHO [46] lower reference limits classify a broad range of infertile men as ‘normal’ where FISH could be indicated. Sperm aneuploidy in normozöospermic men, with a normal somatic karyotype, may be one of the factors related to unexplained male infertility.

A strong association between age and total SDF validates our previous findings [53] and the work of others [77,78,79]. Johnson et al. reported an increase in SDF and a decline in semen quality associated with advancing male age [80]. Germ cell apoptosis during spermatogenesis, which is a normal event, may be less effective in older men, resulting in the release of more DNA-fragmented sperm [81]. It has also been shown that men above 40 years old have significantly higher levels of ROS in their seminal plasma [82]. The group of Plastira et al. found a positive and statistically significant correlation between patient’s age and the percentage of CMA3-stained spermatozoa, suggesting that protamination decreases with age, resulting in looser and more vulnerable chromatin [78]. Due to low numbers and methodological differences, we could not confirm these observations. There is a relative linear correlation between advanced paternal age and sperm aneuploidy [83]. Increasing paternal age, together with alterations in the male endocrinal and reproductive phenotypes [84], leads to the accumulation of DNA fragmentation over years and the decreased capacity of germ cells to repair this damage. This decline in genome integrity might lead to the production of aneuploidy sperm, which translates to increased aneuploidy in embryos [85]. Kaarouch et al. showed that the rate of sperm aneuploidy was significantly higher in men with advanced age (≥40 years) compared to younger men (14% vs. 4%, respectively) [86]. Increased rates of aneuploidy may be associated with arrested spermatogenesis [87].

Male obesity may affect semen parameters [88,89], alter sperm function [90], increase sperm DNA damage [91,92], and induce seminal OS [93]. The underlying mechanism is the adipose tissue that produces pro-inflammatory cytokines, which increase ROS production by leukocytes [94]. Furthermore, the accumulation of adipose tissue within the groin region results in the heating of the testicle, which has been linked with OS and reduced sperm quality [95]. Increasing proportions of men have a high BMI during their reproductive age and these proportions also rise with increasing age, as shown in our results. There was a correlation with BMI and total SDF, probably due to age. Regarding chromatin condensation, the results are conflicting. Some authors [96,97] did not find any association with BMI, while La Vignera et al. found a statistically significant decrease in chromatin condensation in both overweight and obese men when compared to controls [98]. However, none of these studies analyzed chromatin decondensation potential or hypercondensation in relation to the BMI, which makes our study unique. Obese men producing spermatozoa with diminished decondensation potential and greater hypercondensation were also observed in both overweight and obese groups. A link between high paternal BMI and decreased live birth outcomes after assisted reproduction has been reported [99], possibly due to reduced blastocyst development, reduced implantation rates, and higher pregnancy loss [100].

It is estimated that 35% of reproductive-aged males smoke [101], although in our study a lower percentage (23.2%) was noted. Smoking not only affects semen parameters [102,103] but also can impact DNA integrity [104,105]. Smoking could result in an 48% increase in seminal leukocyte concentration and a 107% increase in semen ROS levels [106], while seminal plasma antioxidant levels, on the other hand, are decreased in smokers [107]. Our study could not support this observation as ORP was not affected and there was no significant increase in leukocytes (peroxidase positive leukocytes 0.32 ± 0.9 M/mL in smokers and 0.57 ± 2.2 M/mL in non-smokers; *p* = 0.4818). Conversely, SDF was also not affected. Mostafa et al. indicated that cigarette smoking has detrimental effects on all semen parameters in addition to sperm chromatin condensation [108]. These abnormalities were proportional to the number of cigarettes smoked/day and to the duration of smoking. We could not confirm this observation.

Excessive alcohol consumption causes an increase in systemic OS, as ethanol stimulates the production of ROS, and many alcohol abusers have diets deficient in protective antioxidants [109]. La Vignera et al. observed that excessive ethanol intake was associated with morphologically abnormal spermatozoa, a reduction in spermatogenesis, decreased semen volume, and increased OS [98]. We could not provide conclusive evidence linking alcohol with OS and SDF. However, alcohol intake seemed to help reduce the percentage of hypercondensed chromatin. Although, the numbers analyzed in the alcohol abusers’ group were low.

Use of illicit drugs adversely affects spermatogenesis [110]. Recreational drug use, such as opioids and cannabis abuse, is correlated with high DNA fragmentation in sperm [111]. The effects depend on dosage, duration of usage, and interactions with other drugs [112]. Despite there being few data concerning recreational drug use and OS, studies do show significant adverse effects on semen quality [113]. However, in this study, although drug abuse seemed to reduce total SDF, multivariate regression could not substantiate this finding. The effect on chromatin maturity and stability could not be shown due to the low numbers analyzed.

OS has been linked with extremes of physical activity at both ends of the spectrum. Vigorous activity causes a high muscle aerobic metabolism, creating large amounts of ROS [114]. A sedentary lifestyle and lack of exercise, on the other hand, may increase the pressure force on the testicles and disrupt the intrascrotal temperature regulation, resulting in OS [115,116]. Vital SDF was positively and significantly linked with ORP. Physical activity was non-significantly negatively associated with vital SDF (*p* = 0.053).

If OS represents a relevant clinical issue to male gametes, supplementation with oral antioxidants might improve gamete quality. A plethora of different products have been used clinically with variable results on genome stability to support their use. Table 7 summarizes the studies investigating the effect of oral supplementations on gamete instability. The antioxidant cocktails used influence a finite number of oxy-redox reactions that, according to the dose of antioxidants administered, will be possibly imbalanced towards reductive stress affecting sperm functions [117]. The nutritional supplementation used in our intervention has previously confirmed that the formulation exerts antioxidant modulation (significantly reducing the sperm DNA fragmentation index) and the antioxidant gain (significantly improving the sperm nuclear decondensation index) does not generate any reductive stress [118,119]. Using the same nutritional supplementation, we found that the benefit was more in the vital SDF while the total SDF reacted variably. Whether this imbalance observed in total SDF is due to reductive stress needs to be substantiated in a larger population.

While a healthy diet is certainly conducive to a healthy body and hence a potential association with gamete quality [120] and genome stability, we are a long way from concluding that the supplementation of the diet would be beneficial for male infertility. Large, randomized placebo-controlled trials are required to address this question as concluded in the last Cochrane review of antioxidants for male infertility [121].

In considering the potential implications of our findings, it is appropriate to consider first the methodological strengths. We employed a direct TUNEL assay, and standardized and obtained thresholds to discriminate normal and pathologic conditions. The sensitivity of the TUNEL assay was increased by decompacting the chromatin and adding a live/dead stain to allow the simultaneous assessment of DNA damage and cell viability. Moreover, the assays were performed on fresh semen samples to overcome methodological errors, which might significantly undermine the potential diagnostic value of the test.

Finally, while presenting our results, it is important to recognize and acknowledge any possible limitations. Simultaneous exposure to several lifestyle risk factors impedes the identification of the impact of specific individual factor and may result in synergistic interaction. The issue of self-reporting bias represents a key problem in the assessment of our observational data. Self-reporting data can be affected by social desirability or approval, especially where anonymity and confidentiality in the presence of the spouse/partner cannot be guaranteed at the time of data collection. Moreover, we did not collect data about other risk factors associated with male subfertility (e.g., diabetes, history of mumps, fever).

Whilst the message of ‘no smoking, no alcohol and no drugs’ should be offered as good health advice, our study shows that age and BMI might contribute to sperm genome instability, affecting male fertility. Further large-scale epidemiological studies are required to identify lifestyle risk factors on genome instability affecting male fertility.

## Figures and Tables

**Figure 1 nutrients-14-03155-f001:**
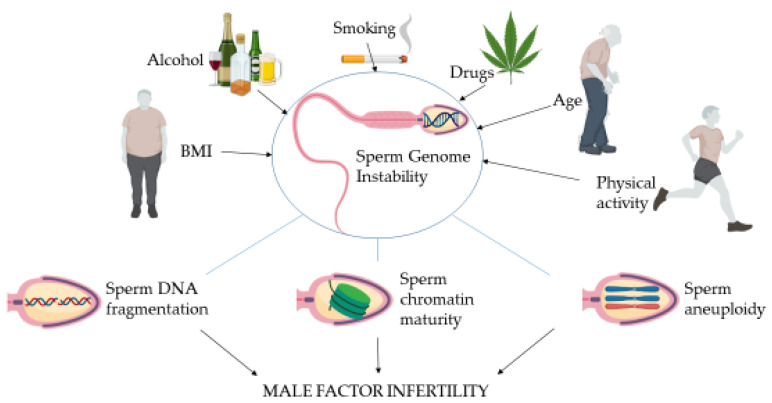
Schematic representation of the lifestyle factors analyzed and methods for genome instability assessment in male factor infertility. Figure created with BioRender.com.

**Figure 2 nutrients-14-03155-f002:**
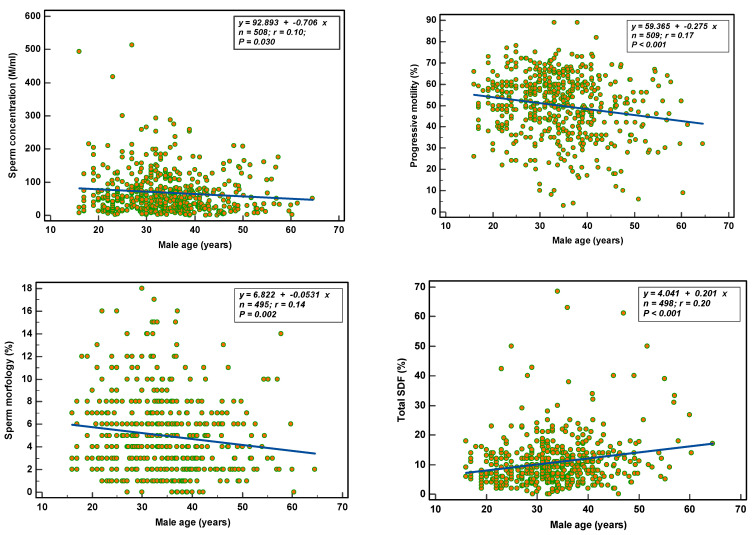
Age-related effects on semen parameters and total SDF.

**Figure 3 nutrients-14-03155-f003:**
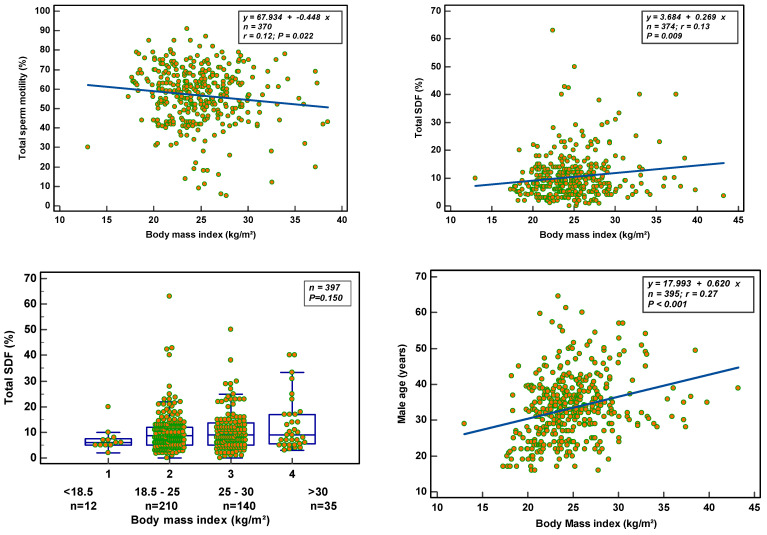
BMI-related effect on semen parameters, total SDF and age.

**Figure 4 nutrients-14-03155-f004:**
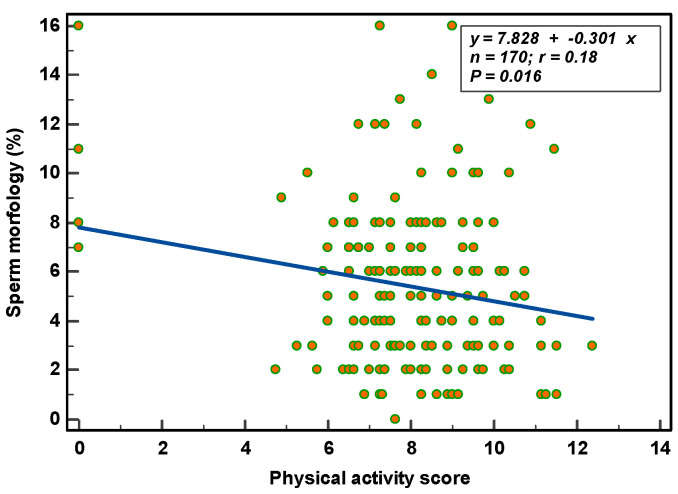
Effects of physical activity on sperm morphology.

**Figure 5 nutrients-14-03155-f005:**
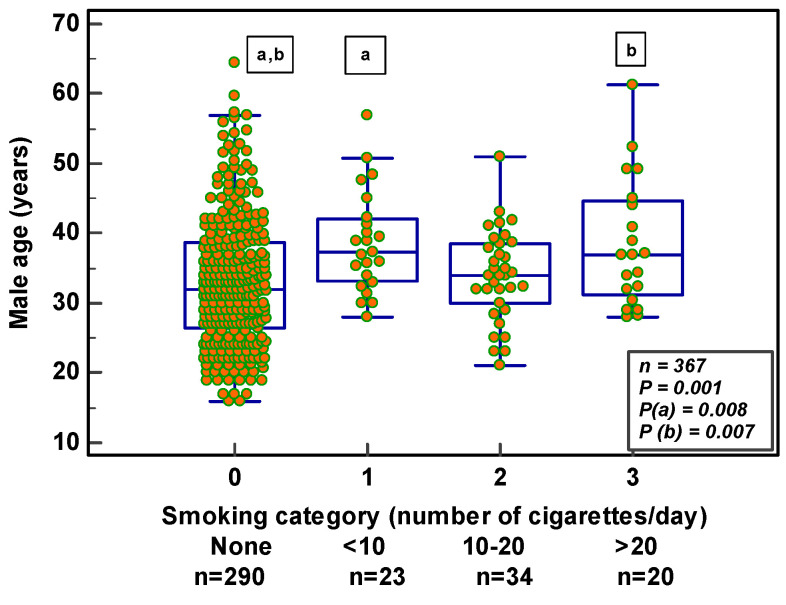
Relation between male age and number of cigarettes smoked per day. Similar letters demonstrate significant differences between the different smoking categories.

**Figure 6 nutrients-14-03155-f006:**
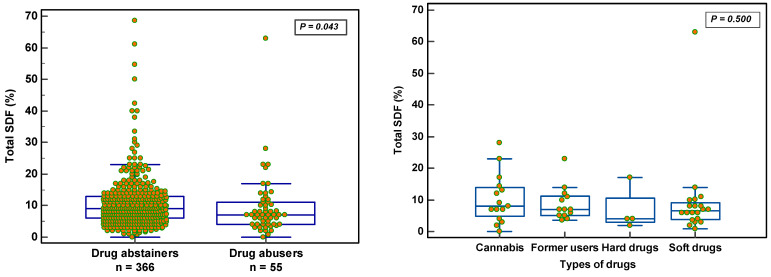
Effect of drug abuse and different types of drugs used on total SDF.

**Figure 7 nutrients-14-03155-f007:**
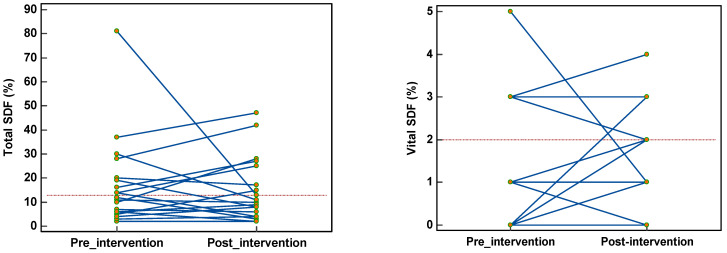
Effect of nutrient supplementation on total and vital SDF.

**Table 1 nutrients-14-03155-t001:** Descriptive statistics of all participants.

Parameters	Numbers	Mean ± SD (Range)
Demographic variables		
Male age at diagnosis (years)	580	33.8 ± 8.9 (18.0–64.5)
Body mass index (kg/m^2^)	440	24.9 ± 3.9 (13.0–43.2)
Smoking		
Non-smokers	343 (77.6%)	
Smokers	99 (22.4%)	
Alcohol		
Abstainers	128 (28.6%)	
Alcohol users	319 (71.4%)	
Drugs		
Abstainers	389 (87.2%)	
Drug users	57 (12.8%)	
Physical activity score	173	8.0 ± 2.0 (0.0–12.4)
ORP (mV/M/mL)	241	3.1 ± 11.9 (−3.7–163.2)

Data are presented as mean ± SD (range) where applicable. ORP = oxidation-reduction potential.

**Table 2 nutrients-14-03155-t002:** Effect of smoking on age, BMI, semen and SDF parameters.

	Non-Smokers	Smokers		*p* Value *
		Current Smokers	Former Smokers	
Male age (years)	33.1 ± 9.2 (16.0–64.5) *n* = 291	36.6 ± 7.8 (21.0–61.3) *n* = 80	37.1 ± 3.6 (32.4–42.9) *n* = 6	<0.001
BMI (kg/m^2^)	24.5 ± 3.7 (13.0–38.5) *n* = 265	26.1 ± 4.1 (19.0–37.4) *n* = 67	25.1 ± 3.2 (20.2–28.7) *n* = 6	0.021
Sperm concentration (M/mL)	70.7 ± 65.5 (0.6–512.5) *n* = 290	60.2 ± 62.7 (0.6–300.0) *n* = 81	40.7 ± 13.8 (26.3–60.0) *n* = 6	0.068
Progressive sperm motility (%)	51.1 ± 12.8 (4.0–89.0) *n* = 290	46.8 ± 16.3 (3.0–75.0) *n* = 81	52.8 ± 8.5 (44.0–68.0) *n* = 6	0.242
Sperm morphology (%)	5.2 ± 3.4 (0.0–18.0) *n* = 283	4.4 ± 3.6 (0.0–17.0) *n* = 79	6.0 ± 2.1 (4.0–9.0) *n* = 6	0.025
Total SDF (%)	10.4 ± 7.9 (0.0–63.0) *n* = 278	9.8 ± 8.4 (0.0–42.3) *n* = 72	9.8 ± 6.5 (4.0–21.0) *n* = 6	0.324
Vital SDF (%)	1.3 ± 1.7 (0.0–14.0) *n* = 278	1.0 ± 1.0 (0.0–4.2) *n* = 72	1.0 ± 0.6 (0.0–2.0) *n* = 6	0.797
ORP (mV/M/mL)	2.6 ± 7.6 (−3.7–57.3) *n* = 107	2.0 ± 3.6 (−0.2–18.2) *n* = 39	1.4 ± 0.8 (0.8–3.0) *n* = 6	0.212

Data are presented as mean ± SD (range). BMI = body mass index; SDF = sperm DNA fragmentation; ORP = oxidation-reduction potential. * (Kruskal–Wallis test).

**Table 3 nutrients-14-03155-t003:** Multiple regression analyses between SDF and other lifestyle parameters.

Parameters	Total SDF		Vital SDF	
	Coefficient (SE)	*p* Value	Coefficient (SE)	*p* Value
Age (years)	0.1950 (0.0564)	<0.001	0.0019 (0.0148)	0.899
BMI (kg/m^2^)	0.1335 (0.1430)	0.352	0.0044 (0.0375)	0.907
smoking	−1.9127 (1.4703)	0.195	−0.5473 (0.3856)	0.158
Alcohol	0.1608 (1.2769)	0.900	0.1617 (0.3349)	0.629
Drugs	−0.1544 (1.4262)	0.914	−0.0325 (0.3741)	0.931
Physical activity score	−0.3168 (0.3384)	0.351	−0.1275 (0.0888)	0.153

SE = standard error; SDF = sperm DNA fragmentation; BMI = body mass index.

**Table 4 nutrients-14-03155-t004:** Multiple regression analyses between ORP and SDF parameters.

Parameters	ORP	
	Coefficient (SE)	*p* Value
Total SDF (%)	−0.2009 (0.1129)	0.077
Vital SDF (%)	1.0999 (0.4291)	0.011

SE = standard error; SDF = sperm DNA fragmentation; ORP = oxidation-reduction potential.

**Table 5 nutrients-14-03155-t005:** Semen, SDF, chromatin parameters and frequency of sperm aneuploidy in the fertile and subfertile groups.

Parameters	Fertile Group	Subfertile Group	*p* Value
Semen parameters	(*n* = 44)	(*n* = 511)	
Sperm concentration (M/mL)	82.3 ± 50.2 (16.7–263.8)	68.8 ± 67.4 (0.6–512.5)	0.006
Total sperm count (M)	288.9 ± 190.8 (21.7–767.3)	233.8 ± 217.3 (1.0–1436.2)	0.013
Progressive motility (%)	57.9 ± 9.1 (34.0–74.0)	50.0 ± 14.4 (3.0–89.0)	<0.001
Total motility (%)	66.4 ± 8.5 (46.0–82.0)	56.6 ± 14.6 (5.0–91.0)	<0.001
Morphology (%)	8.4 ± 4.2 (1.0–22.0)	5.0 ± 3.5 (0.0–18.0)	<0.001
SDF parameters	(*n* = 46)	(*n* = 501)	
Total SDF (%)	10.6 ± 8.6 (1.4–54.6)	10.7 ± 8.5 (0.0–68.6)	0.976
Vital SDF (%)	1.4 ± 1.5 (0.0–6.6)	1.4 ± 2.2 (0.0–25.0)	0.508
Chromatin parameters	(*n* = 10)	(*n* = 65)	
Chromatin condensation (%)	84.5 ± 7.2 (67.0–92.0)	68.0 ± 12.4 (28.9–90.0)	<0.001
Chromatin decondensation (%)	89.9 ± 2.6 (85.0–93.0)	68.0 ± 17.8 (5.4–90.2)	<0.001
Chromatin hypocondensation (%)	7.8 ± 2.6 (5.0–14.0)	9.2 ± 7.0 (1.4–54.6)	0.601
Chromatin hypercondensation (%)	2.1 ± 0.9 (1.0–4.0)	10.8 ± 7.8 (1.6–33.1)	<0.001
Frequency of sperm aneuploidy	(*n* = 20)	(*n* = 203)	
Chromosome 13			
Nullisomy (%)	0.11 ± 0.15	0.16 ± 0.26	0.282
Disomy (%)	0.14 ± 0.11	0.17 ± 0.30	0.294
Chromosome 18			
Nullisomy (%)	0.06 ± 0.09	0.19 ± 0.42	0.061
Disomy (%)	0.12 ± 0.16	0.20 ± 0.45	0.234
Chromosome 21			
Nullisomy (%)	0.08 ± 0.11	0.14 ± 0.17	0.072
Disomy (%)	0.08 ± 0.09	0.16 ± 0.35	0.605
Chromosome X/Y			
Nullisomy (%)	0.30 ± 0.31	0.31 ± 0.48	0.422
Disomy XX (%)	0.15 ± 0.21	0.11 ± 0.16	0.814
Disomy XY (%)	0.10 ± 0.12	0.22 ± 0.66	0.32
Disomy YY (%)	0.07 ± 0.10	0.07 ± 0.15	0.833
Autosomal aneuploidy (%)	0.57 ± 0.34	1.03 ± 1.10	0.04
Sex aneuploidy (%)	0.61 ± 0.45	0.71 ± 0.99	0.779
Diploidy (%)	0.48 ± 0.39	1.17 ± 1.97	0.004

Data are presented as mean ± SD (range). SDF = sperm DNA fragmentation.

**Table 6 nutrients-14-03155-t006:** Semen and SDF parameters in the subfertile group before and after intervention.

Parameters	Before Intervention	After Intervention	% Difference	*p* Value *
Semen volume (mL)	3.3 ± 1.3	3.4 ± 1.9	0.2	0.8457
Sperm concentration (M/mL)	57.9 ± 63.2	64.0 ± 67.1	6.1	0.7706
Total count (M/ejaculate)	179.4 ± 192.6	212.3 ± 291.2	32.8	0.9033
Progressive motility (%)	36.4 ± 18.9	46.5 ± 16.8	10.1	0.0844
Total motility (%)	46.1 ± 18.5	56.0 ± 17.3	9.9	0.0596
Sperm morphology (%)	3.6 ± 2.6	3.9 ± 2.9	0.3	0.7311
Total SDF (%)	18.7 ± 18.1	14.9 ± 12.9	−3.8	0.4973
Vital SDF (%)	1.5 ± 1.5	1.6 ± 1.1	0.1	0.4666

Data are presented as mean ± SD; SDF = sperm DNA fragmentation; * Mann–Whitney test.

**Table 7 nutrients-14-03155-t007:** Studies investigating the effect of oral supplementations on SDF.

Study	Supplement/Day	Duration	Study Design and Patient Population	SDF Assay	Study Results
Fraga et al. [122]	Vitamin C (250 mg) Depletion to 5 mg/day, Repletion to 250 mg/day	15 weeks	Prospective, observational study; 24 normal volunteers	8-OHdG	DNA damage increased by 91% upon depletion due to reduced seminal ascorbic acid, 36% could be restored by repletion
Kodama et al. [123]	GSH (400 mg) Vitamin C (200 mg) Vitamin E (200 mg)	2 months	Prospective, observational study; 14 infertile men	8-OHdG	Modest decrease in 8-OHdG levels from 1.5 ± 0.2 to 1.1 ± 0.1/105 deoxyguanosine (*p* < 0.05)
Greco et al. [124]	Vitamin C (1000 mg) Vitamin E (1000 mg)	2 months	Prospective, observational; 38 infertile males with DFI > 15%	TUNEL	29/38 responded with a decrease in SDF from 24.0 ± 7.9 to 8.2 ± 4.3 (*p* ≤ 0.001) while 9/38 showed no difference in SDF values from 25.1 ± 8.5 to 23.8 ± 9.2%
Greco et al. [125]	Vitamin C (1000 mg) Vitamin E (1000 mg)	2 months	Randomized placebo-controlled study; 64 infertile males with DFI > 15%	TUNEL	Decrease in SDF from 22.1 ± 7.7% to 9.1 ± 7.2 (*p* < 0.001) in the treatment group, but not in the placebo group (from 24.4 ± 7.8 to 22.9 ± 7.9)
Menezo et al. [118]	Vitamin C (400 mg) Vitamin E (400 mg) Zinc (33 mg) Selenium (80 µg) β-carotene (18 mg)	90 days	Double-centered, observational study; 58 males with DFI >15%	SCSA	DFI decreased from 32.4% to 26.2% (*p* < 0.001) but, sperm decondensation increased from 17.5% to 21.5% (*p* < 0.001)
Tremellen et al. [126]	MenevitR: zinc (25 mg) Vitamin C (100 mg) Vitamin E (400 IU) Lycopene (6 mg) Garlic oil (33 µg) Selenium (26 µg) Folic acid (500 µg)	3 months	Double-blind randomized, controlled study; 60 with severe male factor infertility	TUNEL	DNA damage reduced from 37.9% to 33.3%; but from 40.03% to 32.0% in controls
Piomboni et al. [127]	Beta-glucan (20 mg) fermented Papaya (50 mg) Lactoferrin (97 mg) Vitamin C (30 mg) Vitamin E (5 mg)	3 months	Prospective study; 36 men with leukocytospermia and 15 controls	SCSA	No significant decrease in SDF in the control (15.8 ± 6.7 to 16.1 ± 5.4) and treatment (16.7 ± 8.0 to 14.4 ± 6.0) groups
Omu et al. [128]	Group1: zinc (400 mg) Group 2: zinc (400 mg) + vitamin E (20 mg) Group 3: zinc (400 mg) + vitamin E (20 mg) + vitamin C (10 mg)	3 months	Randomized placebo-controlled study; 45 men with asthenozoospermia, 37 treatment group (group 1 = 11; group 2 = 12; group 3 = 14), 8 placebo group	SCSA	Zinc supplementation resulted in significantly lower DFI (14–29%, *p*< 0.05) compared to zinc deficiency.
Tunc et al. [129]	MenevitR: as above	3 months	Prospective, observational study; 50 males with oxidative stress	TUNEL	SDF levels dropped from 22.2% to 18.2% (*p* = 0.002) and sperm DNA protamination improved from 69.0% to 73.6% (*p*< 0.001)
Vani et al. [130]	Vitamin C (1000 mg) 5 consecutive days in a week	3 months	Prospective, comparative study; 120 men exposed to lead, and 120 healthy human subjects	Comet	Decrease in alkaline-labile sites and mean tail length of the comet when compared to the control group (*p* < 0.01)
Abad et al. [131]	L-carnitine (1500 mg) Coenzyme Q10 (20 mg) Vitamin C (60 mg) Vitamin E (10 mg) Vitamin B9 (200 µg) Vitamin B12 (1 µg) Zinc (10 mg) Selenium (50 µg)	3 months	Prospective, observational study; 20 asthenoterato-zoospermic infertile males	SCD	DNA damage reduced from 28.5% ± 14.97% to 20.12% ± 8.26% (*p* = 0.004) DNA degraded sperm also reduced from 7.32% ± 4.12% to 5.66% ± 3.21% (*p* = 0.04)
Dattilo et al. [119]	CondensylR: Opuntia fig fruit (100 mg) Quercetin (0.05 mg) Betalain (0.001 mg) Vitamin B2 (1.4 mg) Vitamin B3 (16 mg) Vitamin B6 (1.4 mg) Vitamin B9 (400 µg) Vitamin B12 (2.5 µg) Vitamin E (12 mg) n-acetyl-cysteine (250 mg)	4 months	Prospective, observational study; 84 infertile men	TUNEL	DFI decreased from 29.7% to 23.1% (*p* < 0.001); sperm nuclear decondensation index decreased from 40.1% to 36.3% (*p* < 0.001)
Gual-Frau et al. [132]	L-carnitine (1500 mg) Vitamin C (60 mg) Coenzyme Q10 (20 mg) Vitamin E (10 mg) Zinc (10 mg) Vitamin B9 (200 μg) Selenium (50 μg) Vitamin B12 (1 µg)	3 months	Prospective, observational study; 20 infertile men with grade I varicocele	SCD	After treatment, an average relative reduction of 22.1% in SDF (*p* = 0.02) and 31.3% fewer highly degraded sperm cells (*p* = 0.07) were observed
Martínez-Soto et al. [133]	Docosahexaenoic acid (1500 mg)	10 weeks	Randomized, double-blind, placebo-controlled, parallel-group study	TUNEL	Decrease in SDF values (−17.2 ± 2.8%, *p* < 0.001) in treatment group vs. (+11.2 ± 1.9%, *p* > 0.05) in the placebo group
Barekat et al. [134]	N-acetyl-L-cysteine (NAC; 200 mg) three times daily	3 months	Randomized controlled trial; 35 infertile men with varicocele, subjected to varicocele repair; 20 control group; 15 treatment group	TUNEL	Improvement in sperm chromatin integrity in men subjected to varicocelectomy receiving NAC post-surgery compared to those who did not (11.8% ± 2.01 vs. 4.7% ± 1.3, *p* < 0.01)
Stenqvist et al. [135]	Vitamin C (30 mg) Vitamin E (5 mg) Vitamin B12 (0.5μg) l-carnitine (750 mg) coenzyme Q10 (10 mg) Folic acid (100 μg) Zinc (5 mg) Selenium (25 μg)	3 and 6 months	Randomized, double-blind, placebo-controlled study; 37 treatment group; 40 placebo group	SCSA	No significant decrease in DFI both in the placebo and treatment groups, after 3 and 6 months of supplementation

DFI = DNA fragmentation index; GSH = glutathione; 8-OHdG = 8-hydroxy-2-deoxyguanosine; TUNEL = Tdt (terminal deoxynucleotidyl transferase)-mediated dUDP nick-end labelling; SCSA = sperm chromatin structure assay; SCD = sperm chromatin dispersion test.

## Data Availability

The data presented in this study are available from the corresponding author on a reasonable request.

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
