# Peer review of "Sperm as a Carrier of Genome Instability in Relation to Paternal Lifestyle and Nutritional Conditions"

_nutrients, 2022, doi:10.3390/nu14153155_

Round 1

Reviewer 1 Report

Dear authors,

Thank you for inviting me as a reviewer of this article.

I thoroughly checked the methods and values of the manuscript.

I have following suggestions:

1. "Table 1. Descriptive statistics of all participants" and "Table 5. Semen, SDF, chromatin parameters and sperm aneuploidy in the fertile and 434 subfertile groups."

The parameters are not aligned properly.

2. In the ‘2.9. Statistical analysis’ section, there are no references for selecting your current methodology. Please justify yourself about selecting those methods, or refer to some articles on how to select proper statistical methods. 

Unpaired Student T-test was used in case of normal distribution and the 286 Mann-Whitney test was used in case the data were not normally distributed. 287 Differences in continuous variables between 3 or more groups were assessed 288 using the ANOVA and Kruskall-Wallis tests. [1]

[1] https://doi.org/10.54724/lc.2022.e1

OR

We used Unpaired Student T-test because ~~~~. Mann-Whitney test was selected due to ~~~~~

3. If some of the variable is not normally distributed, I suggest you to list up those variables.

Thank you.

Author Response

Manuscript ID: nutrients-1825191

Title: Sperm as a carrier of genome instability from paternal lifestyle and nutritional conditions

First of all we would like to express our deep sense of gratitude to the reviewer for the time and energy spent in reading and reviewing our study. We have thoroughly revised and modified the manuscript where necessary and we hope that the revised edition will meet the expectations of the reviewer.

All revisions made to the manuscript have been done so using the “Track Changes” function. We have also carried out a spelling and grammatical check for the English language and the style modified where required. The relevancy and the correctness of the references used have also been controlled.  

We do regret to draw attention to the fact that the template of the submitted manuscript was affected during the submission process. The page ruler was modified from page 2 onwards, as a result of which the alignment of the figures and tables had been affected. These have been concomitantly rectified as far as possible.

Response to reviewer 

Comment 1:  "Table 1. Descriptive statistics of all participants" and "Table 5. Semen, SDF, chromatin parameters and sperm aneuploidy in the fertile and 434 subfertile groups." - The parameters are not aligned properly.

Answer: Thank you for this important observation. Table 1 (page 7, line 311) and Table 5 (page 13, line 444) are now appropriately aligned as suggested.

Comment 2: In the ‘2.9. Statistical analysis’ section, there are no references for selecting your current methodology. Please justify yourself about selecting those methods, or refer to some articles on how to select proper statistical methods. 

Answer: First of all we would like to acknowledge the reviewer's suggestion of a reference concerning statistical methods. Concomitantly, we have used the article as suggested by the reviewer and made the following constructive modification (pages 7, lines 287-290):

Unpaired Student T-test was used in case of normal distribution and the Mann-Whitney test was used in case the data were not normally distributed. Differences in continuous variables between 3 or more groups were assessed using the ANOVA and Kruskall-Wallis tests. [59]

Comment 3: If some of the variable is not normally distributed, I suggest you to list up those variables.

Answer: The authors appreciate the reviewers suggestion and have made the following modifications (page 7, lines 293-297):

With an exception of chromatin condensation data, where an unpaired student t-test was conducted, all other chromatin, semen and SDF parameters “rejected Normality” (P<0.05) for which the Mann-Whitney test was used to assess differences in continuous variables between two groups.

We thank you in advance for considering our work and look forward to hearing from you.

Reviewer 2 Report

The manuscript numbered 1825191 deals with the influence of lifestyle and nutritional factors on the genome instability of sperm.

The topic of this paper may be of interest for the journal’ audience. Manuscript raise the most recent scientific matters and provide proper explanation of them. The paper is well written, the need of the research is enough justified, experiment was well thought out, planned and executed. Materials and methods are exhaustively described, used methods are novel and well-chosen to achieve the main aim of the study. Obtained results were sufficiently described discussed, discussion are explanatory and informative.

Author Response

Manuscript ID: nutrients-1825191

Title: Sperm as a carrier of genome instability from paternal lifestyle and nutritional conditions

First of all we would like to express our deep sense of gratitude to the reviewer for the time and energy spent in reading and reviewing our study. We have thoroughly revised and modified the manuscript where necessary and we hope that the revised edition will meet the expectations of the reviewer.

All revisions made to the manuscript have been done so using the “Track Changes” function. We have also carried out a spelling and grammatical check for the English language and the style modified where required. The relevancy and the correctness of the references used have also been controlled.  

We do regret to draw attention to the fact that the template of the submitted manuscript was affected during the submission process. The page ruler was modified from page 2 onwards, as a result of which the alignment of the figures and tables had been affected. These have been concomitantly rectified as far as possible.

Response to reviewer 

Comment 1: I am very happy reviewing this manuscript. I find it succinct and addressing important matters in reproductive medicine. The laboratory technical parts I can take in are adequate and well described.

Answer: Thank you for this positive response. We are overwhelmed and flattered that the reviewer appreciates the main points of our work and we humbly acknowledge this immense motivation.

We thank you in advance for considering our work and look forward to hearing from you.

This manuscript is a resubmission of an earlier submission. The following is a list of the peer review reports and author responses from that submission.